# Use of Coronary Computed Tomography Angiography to Screen Hospital Employees with Cardiovascular Risk Factors

**DOI:** 10.3390/ijerph18105462

**Published:** 2021-05-20

**Authors:** Po-Yi Li, Ru-Yih Chen, Fu-Zong Wu, Guang-Yuan Mar, Ming-Ting Wu, Fu-Wei Wang

**Affiliations:** 1Department of Family Medicine, Kaohsiung Veterans General Hospital, Kaohsiung 813, Taiwan; ventomaple@hotmail.com (P.-Y.L.); rrychen@vghks.gov.tw (R.-Y.C.); 2Department of Radiology, Kaohsiung Veterans General Hospital, Kaohsiung 813, Taiwan; fzwu@vghks.gov.tw (F.-Z.W.); mtwu@vghks.gov.tw (M.-T.W.); 3Division of Cardiology, Department of Internal Medicine, Kaohsiung Veterans General Hospital, Kaohsiung 813, Taiwan; gymar@vghks.gov.tw

**Keywords:** coronary computed tomography angiography (CCTA), screen, Framingham Risk Score, Coronary Artery Disease Reporting and Data System (CAD-RADS), coronary artery disease

## Abstract

The objective of this study was to determine how coronary computed tomography angiography (CCTA) can be employed to detect coronary artery disease in hospital employees, enabling early treatment and minimizing damage. All employees of our hospital were assessed using the Framingham Risk Score. Those with a 10-year risk of myocardial infarction or death of >10% were offered CCTA; the Coronary Artery Disease Reporting and Data System (CAD-RADS) score was the outcome. A total of 3923 hospital employees were included, and the number who had received CCTA was 309. Among these 309, 31 (10.0%) had a CAD-RADS score of 3–5, with 10 of the 31 (32.3%) requiring further cardiac catheterization; 161 (52.1%) had a score of 1–2; and 117 (37.9%) had a score of 0. In the multivariate logistic regression, only age of ≥ 55 years (*p* < 0.05), hypertension (*p* < 0.05), and hyperlipidemia (*p* < 0.05) were discovered to be significant risk factors for a CAD-RADS score of 3–5. Thus, regular and adequate control of chronic diseases is critical for patients, and more studies are required to be confirmed if there are more significant risk factors.

## 1. Introduction

Heart disease, including coronary artery disease, is one of the 10 most common causes of death each year and is thus a crucial issue. Approximately 11% of deaths in Taiwan during 2019, the year for which the most recent statistics are available, were caused by heart disease, with this disease being the second leading cause of death in both sexes [1]. In 2018, approximately 23% of deaths in the United States were caused by heart disease, making it the biggest killer [2]. The newest statistics released by the World Health Organization, which are for 2019, report heart disease to be the most common cause of mortality worldwide, accounting for 16% of deaths [3]. An ischemic heart disease episode places large burdens and causes great challenges to not only a patient’s body, mind, and spirit but also their family’s finances, stress, and caregiving. According to estimates made in 2019, ischemic heart disease is responsible for 7.1% of disability-adjusted life years and 10.3% of years of life lost, the second highest percentages of any cause [4].

Numerous studies have revealed an association between the risk of coronary heart disease and long working hours [5,6], especially ≥55 working hours per week [7,8], job strain [9,10,11], and shift work [12,13,14]. Compared with those working in other occupations, hospital employees more frequently experience emergencies, life-threatening problems, distress, and emotional fluctuation due to issues with patients and their families; these experiences cause substantial strain and elevate coronary heart disease risk.

Coronary atherosclerosis, with which ischemic heart disease has an association, was previously evaluated using invasive coronary plaque imaging techniques, including optical coherence tomography and intravascular ultrasound [15,16]. The noninvasive imaging technique, coronary computed tomography angiography (CCTA), which does not require fasting or hospitalization, has recently been developed for evaluating coronary atherosclerosis [17,18], and a standardized report, the Coronary Artery Disease Reporting and Data System (CAD-RADS), has been produced [19,20]; CCTA is especially useful for assessing asymptomatic individuals with risk factors [21,22,23]. Some studies have used CCTA to investigate the relationship between coronary artery disease and shift work in a specific group [24] and to compare it with myocardial perfusion imaging in patients with high occupational risk [25].

The Framingham Risk Score [26], a 10-year risk of myocardial infarction and death rate, is a scoring system that was published in 1998 [27]. The algorithm of multivariate risk functions by sex that incorporated age, total cholesterol, high-density lipoprotein, systolic and diastolic blood pressure, smoking, and clinical diagnosis of hypertension and diabetes mellitus was derived. The Framingham Risk Score has been validated and can be applied to various ethnic groups [28,29,30].

To ensure that hospital employees have a healthy work environment and to determine how CCTA should be used for early detection and treatment of coronary artery stenosis, the researchers of this study considered the existing regular health examinations and personal history questionnaires employed in our hospital and arranged CCTA screening for employees with cardiovascular risk factors. The objective was to intervene early to minimize the potential damage of ischemic heart disease.

## 2. Methods

All employees in our hospital were included in this study. The Framingham Risk Score was calculated using data from employees’ regular health examinations, their medical history, and data from personal history questionnaires. The data included age; smoking history; systolic and diastolic blood pressure; high-density lipoprotein and total cholesterol levels; and clinical diagnosis of diabetes mellitus, hypertension, hyperlipidemia, and heart disease. The clinical diagnosis of diabetes mellitus, hypertension, and hyperlipidemia is based on employees’ medical history, existing drug control, or a new diagnosis made by their family physician after tracking abnormal data that have been found in regular health examinations. The definition for diabetes mellitus is based on the criteria of the American Diabetes Association [31]; for hypertension, it is based on the criteria of the European Society of Cardiology and the European Society of Hypertension [32]; for hyperlipidemia, it is based on the criteria of the American Heart Association [33]; and heart disease consists of coronary artery disease, valvular heart disease, and arrhythmia. Smoking habit is categorized into the following groups: current smoker, ex-smoker, and never smoker; ex-smoker is defined as smoke free for at least 6 months.

CCTA was offered for further evaluation of coronary artery stenosis and plaque to those with a 10-year risk of myocardial infarction or death of >10%, with priority given to those with a higher risk. Regarding the CCTA protocol, a beta-blocker was used if the employee’s heart rate was >65 bpm until it had dropped to <65 bpm. The employee lay down, an 18–20 gauge intravenous catheter was inserted, and electrocardiograms (ECGs) and a respiratory monitoring patch were used during the entire examination. Contrast medium was injected, and scanning was performed while the employee held their breath. After the examination, the employee rested for 20 min to ensure that their heart rate was stable and that they had no abnormal reactions.

The CCTA report, the CAD-RADS, was employed as an outcome. Employees were categorized into groups with CAD-RADS scores of 0 (no coronary stenosis), 1–2 (<50% maximal coronary stenosis), and 3–5 (≥50% maximal coronary stenosis).

Participants with CAD-RADS scores of 1–2 were referred to their family physician for follow-up and education, whereas those with CAD-RADS scores of 3–5 were referred to a cardiologist for follow-up and further treatment. Those with a 10-year risk of myocardial infarction or death of >10% were followed up by personnel from the Department of Occupational Safety and Health, with regular encouragement of exercise and health talk for hypertension, diabetes mellitus, hyperlipidemia, cancer, smoking, and obesity.

In risk factors, the cut-off of age is based on the retirement age of high-risk occupations, which is set at ≥55 years in Article 54 of Labor Standards Law in Taiwan [34]; the cut-off of BMI is based on the definition of obesity as ≥27 kg/m^2^ in Taiwan [35]. Regarding BMI, underweight is set at <18.5 kg/m^2^, normal weight is set at ≥18.5 kg/m^2^ and <24 kg/m^2^, and overweight is set at ≥24 kg/m^2^ and <27 kg/m^2^.

The Institutional Review Board of Kaohsiung Veterans General reviewed and approved this study (VGHKS17-CT10-05). The employees receiving CCTA signed consent forms after being well informed about the project.

IBM SPSS Statistics, version 20, was used for statistical analysis. The mean ± standard deviation is used to detail distributions. Continuous variables were assessed using Student’s test, whereas discrete variables were evaluated using the chi-squared or Fisher’s exact test. The data were verified by the Kolmogorov–Smirnov test, which assumes a normal distribution or satisfies the sample size guidelines for non-normal data prior to parametric statistical analysis. Significant coronary arterial stenosis risk factors were identified through multivariate logistic regression. A *p* value of < 0.05 was considered to denote statistical significance.

## 3. Results

From January 2017 to December 2020, a total of 3923 hospital employees were included in this study. The number of employees with a 10-year risk of myocardial infarction or death of >10% was 430; the risk level was intermediate in 361 participants and high in 69 participants. Those with a higher Framingham Risk Score had priority for CCTA, and at the time of writing, the number of employees who had undergone CCTA was 309–252 at the intermediate-risk level and 57 at the high-risk level. The characteristics of these 309 hospital employees are listed in Table 1. All employees who have diabetes mellitus have type 2 diabetes mellitus. Refusal to participate was due to resignation in 9 instances, transfer to another hospital in 3, rejection by the employee in 36, and cancellation because of a clinical problem in 4.

Among the included employees, 31 (10.0%, 31/309) had CAD-RADS scores of 3–5 (≥50% maximal coronary stenosis) and were referred to a cardiologist for further evaluation; 10 of these 31 (32.3%) employees required cardiac catheterization. In addition, 161 employees (52.1%) had CAD-RADS scores of 1–2 (<50% maximal coronary stenosis) and were referred to their family physician for advisory and education. Finally, 117 employees (37.9%) had a CAD-RADS score of 0 and were recommended to control their risk factors (Figure 1).

Additionally, 2 employees were accidentally discovered to have early-stage lung cancer, and 15 employees were discovered to have benign lung lesions and were followed up regularly at an outpatient department; 1 of them was subsequently found to have early-stage lung cancer.

In a univariate analysis, employees older than 55 years were identified as having a higher risk of having a CAD-RADS score of 3–5 (significant coronary artery stenosis; odds ratio (OR): 2.954, *p* = 0.005). Hypertension (OR: 2.818, *p* = 0.008), hyperlipidemia (OR: 2.804, *p* = 0.007), and a high risk (Framingham Risk Score >20%) compared with an intermediate risk (10–20%; OR: 2.340, *p* = 0.041) were also significant risk factors for having a CAD-RADS score of 3–5 (Table 2). In multivariate logistic regression, only age of ≥55 years (OR: 2.716, *p* = 0.013), hypertension (OR: 2.287, *p* = 0.039), and hyperlipidemia (OR: 2.635, *p* = 0.014) were significant risk factors for a CAD-RADS score of 3–5 (Table 3).

## 4. Discussion

In Taiwan, the cost of basic regular health examinations for employees is covered by their company. Further and costly examinations are covered only for senior supervisors and older employees. However, based on the concept that every employee is an important member of the organization, this is the first study to include all hospital employees in screening for costly CCTA in Taiwan. The most beneficial use of CCTA for screening is critical.

Accumulated evidence from studies employing CCTA assessments indicates that the following are independent risk factors for coronary atherosclerosis: age [36], diabetes mellitus [37], hypertension [38], smoking [39,40], chronic kidney disease [41], hyperlipidemia [42], and familial hypercholesterolemia [43]. One study revealed that no single risk factor incurred a significantly higher risk of significant artery stenosis, with a combination of risk factors being required [44].

In this study, before a multivariate analysis was performed, the significant risk factors of significant coronary artery stenosis were discovered to be an age of ≥55 years, hypertension, hyperlipidemia, and a Framingham Risk Score of >20%. In the multivariate logistic regression, only age of ≥55 years, hypertension, and hyperlipidemia were discovered to be associated with a higher risk of significant coronary artery stenosis. This finding is different from those in previous reports in which diabetes mellitus and smoking were found to incur a higher cardiovascular risk and be associated with a higher prevalence of asymptomatic coronary artery disease. Hyperlipidemia causes foam cells to accumulate and the intima to thicken focally, leading to fatty streaks [45]. As the disease progresses, the fatty streaks become sites of atherosclerosis; lipids, connective tissue, advanced lesions with a necrotic lipid-rich core, and ultimately calcified regions accumulate in smooth muscle cells [46]. Thus, as well as being an independent risk factor for coronary atherosclerosis [42], hyperlipidemia is directly associated with coronary heart disease [47] and mortality [48,49]. The potential pathogenesis leading to atherosclerosis induced by hypertension has marked similarities to those associated with hyperlipidemia. Both conditions have pro-inflammatory effects, resulting in the adherence of leukocytes to the endothelial surface, accumulation of macrophages in the intima, and enhanced synthesis and deposition of connective tissue components in the vessel wall [50]. One study revealed that hypertension was associated with an increased lifetime risk of overall cardiovascular disease of 63.3% compared to 46.1% for patients with normal blood pressure [51]. In 2015, of the risk factors ranked by attributable ischemic heart disease burden in the former Soviet Union and satellite countries, high blood pressure and high cholesterol ranked as the first and second leading risk factors [52], indicating that adequate control of chronic disease is crucial for patients.

Two employees were accidentally discovered to have lung nodule lesions and were referred to the Division of Chest Surgery, with early-stage lung cancer being diagnosed after surgery. One employee was found to have a benign lung lesion and was followed up regularly at an outpatient department; the employee had a progressive lung lesion, and early-stage lung cancer was diagnosed after surgery. For these employees, early detection and treatment were critical, and they minimized the potential damage of the lesions.

Different approaches should be considered in the future. To determine the appropriate use of CCTA for screening, the Framingham Risk Score was used in our study to indicate the risk of coronary artery disease; in further research, the atherosclerotic cardiovascular disease model, least absolute shrinkage and selection operator [53], or another model could be considered. Regarding outcomes, some studies have employed coronary artery calcium as a surrogate variable of coronary atherosclerotic burden [54,55]; this is different from our study, which used CAD-RADS as the outcome of coronary artery stenosis.

Regarding the use of imaging screens for asymptomatic coronary artery disease remains controversial in both the American College of Cardiology/American Heart Association (ACC/AHA) and the European Society of Cardiology (ESC) guidelines. According to the ACC/AHA guideline, the coronary artery calcium score is considered an appropriate option for asymptomatic patients as an initial evaluation with an estimated >5% 10-year risk, diabetes mellitus, familial hyperlipidemia, or family history of early coronary artery disease; CCTA is suitable for symptomatic patients as an initial evaluation without any history of coronary artery disease. According to the ESC guideline, the calcium score may be considered a risk modifier in the cardiovascular disease risk assessment of asymptomatic patients at moderate risk (IIb); screening of coronary artery disease with CCTA may be considered (IIb) [56]. There is a concern regarding the radiation exposure of CCTA, which is higher than that of the coronary artery calcium score. However, with recently introduced, new CT technologies, the radiation dose for CCTA has been significantly reduced. We adapted the newer CT scanner and 40% of adaptive statistical iterative reconstruction, following the rule of “as low as reasonably achievable”; as a result, the radiation dose was reduced by approximately 30% compared to that of conventional CT. This makes CCTA more advantageous, significantly reducing the difference of radiation dose between CCTA and the coronary artery calcium score, which can offer detailed imaging to provide better analysis and prognostic information. Some studies have demonstrated the benefits of CCTA in asymptomatic patients, including those with diabetes mellitus and non-calcified plaque, and even in young adults [21,57,58], while the coronary artery calcium score is considered an appropriate option by the guidelines.

This study was designed because two asymptomatic senior supervisors in our hospital, who were classified to intermediate-risk level by Framingham Risk Score, both experienced a new cardiovascular event during the same year, and because hospital employees are at greater risk of long working hours, job strain, and shift work, which increases the risk of coronary heart disease. The coronary artery calcium score is a measurement of the amount of calcium in the walls of the arteries that supply the heart muscle, which shows a significant association with the medium- or long-term occurrence of major cardiovascular events. However, the coronary artery calcium score cannot represent the real plaque or stenosis of the coronary artery. The presence of coronary artery stenosis, which was evaluated by CCTA, does not mean that the patient will have a heart attack, but this report still represents the actual stenosis of the coronary artery. We therefore designed this study to offer CCTA to those whose Framingham Risk Score indicated an intermediate- or high-risk level.

This study had some limitations. First, the Framingham Risk Score predicts approximately 65% of cardiovascular incidents but also results in a substantial number of false positives. The overall true-to-false positive ratio is approximately 1:19 [59]. Thus, the Framingham Risk Score may misclassify cardiovascular events. For example, when calculating the Framingham Risk Score, only a recorded diagnosis of diabetes mellitus was considered; details such as time since diagnosis and favorable diabetes control may influence how severe coronary atherosclerosis is. Additionally, the blood pressure score was determined by the time of the regular health examination, and the cardiovascular risk may have been overestimated because of white coat hypertension. The only smoking variables were current smoker, ex-smoker, or never smoker; details such as exact smoking quantity, the number of years smoking, and history of smoking may have also influenced the outcome. Second, the number of employees with a 10-year risk of myocardial infarction or death of >10% was 430, and the number who had undergone CCTA at the time of writing was 309 (71.9%). Employees with a higher risk had a higher priority for CCTA, which may have influenced the statistical results, such as the prevalence of diabetes mellitus and hypertension; this indicates that the present data cannot be regarded as representative of the whole population. However, the present findings demonstrate the actual clinical situation and match the purpose of early detection and treatment to minimize the potential damage to employees. Third, a difficult working environment—including long working hours [5,6], especially 55 working hours or more per week [7,8], job strain [9,10,11], and shift work [12,13,14]—has been shown to increase coronary heart disease risk. However, these factors were not included in our study, and whether a statistically significant difference existed could not be determined. Collecting these data to more comprehensively identify risk factors should be considered in the future. Fourth, the CCTA report only indicated the degree to which an employee had coronary artery stenosis and did not indicate whether actual cardiovascular events had occurred. Finally, this study was conducted in only a single center and involved a small total number of participants; more clinical studies are required to evaluate the applicability of CCTA screening to the general population.

## 5. Conclusions

Among the hospital employees with Framingham Risk Scores of myocardial infarction or death of >10%, 10.0% (31/309) had significant coronary artery stenosis (CAD-RADS score of 3–5), and 32.3% (10/31) required further cardiac catheterization. Age of ≥55 years (*p* < 0.05), hypertension (*p* < 0.05), and hyperlipidemia (*p* < 0.05) were discovered in the multivariate logistic regression to be significant risk factors for significant coronary artery stenosis (CAD-RADS score of 3–5). Thus, regular and adequate control of chronic diseases is crucial, and more related studies are required to confirm whether there are more significant risk factors.

## Figures and Tables

**Figure 1 ijerph-18-05462-f001:**
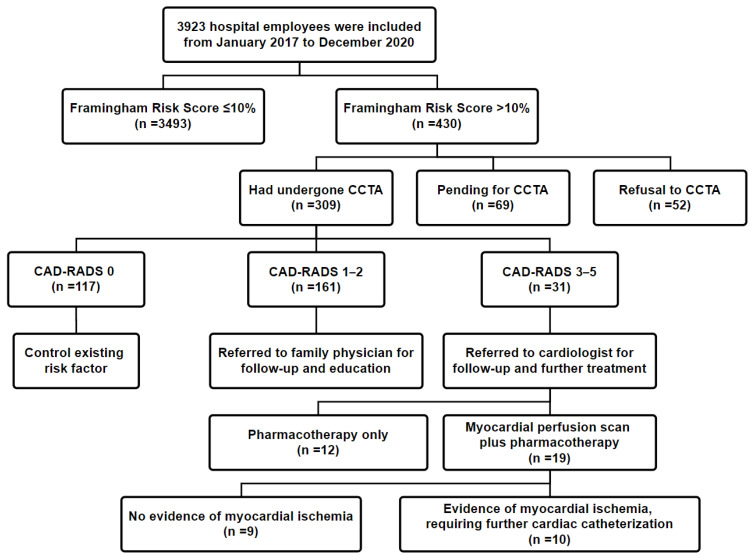
Flowchart for enrollment of the study population. Abbreviation: CCTA: coronary computed tomography angiography (CCTA); CAD-RADS: Coronary Artery Disease Reporting and Data System; PTCA: percutaneous transluminal coronary angioplasty.

**Table 1 ijerph-18-05462-t001:** Baseline data of 309 employees undergoing coronary computed tomography angiography.

Factors	*n* = 309 (%)
Male/female	184 (59.5)/125 (40.5)
Age (y/o) (mean ± S.D.)	52.5 ± 5.6
BMI (kg/m^2^) (mean ± S.D.)	26.7 ± 3.9
eGFR (mL/min/1.73 m^2^) (mean ± S.D.)	78.73 ± 13.99
Diabetes mellitus (+/−)	45 (14.6)/264 (85.4)
Hypertension (+/−)	119 (38.5)/190 (61.5)
Hyperlipidemia (+/−)	101 (32.7)/208 (67.3)
Smoking (current/ex/never -smoker)	21 (6.8)/11 (3.6)/277 (89.6)
Alcohol (+/−)	68 (22.0)/241 (78.0)
HBV or HCV (+/−)	31 (10.0)/278 (90.0)
Heart disease (+/−)	16 (5.2)/293 (94.8)
CVA (+/−)	5 (1.6)/304 (98.4)
10-year risk of myocardial infarction or death rate	
Intermediate-risk level (10–20%)	252 (81.6)
High-risk level (>20%)	57 (18.4)
CAD-RADS	
0	117 (37.9)
1–2	161 (52.1)
3–5	31 (10.0)

**Table 2 ijerph-18-05462-t002:** Prevalence of CCTA-detected significant coronary stenosis in various risk factor subgroups (*n* = 309).

Risk Factor	Total No. of Patients (%)	No. of Patients with Significant Coronary Stenosis	*p*	OR (95% CI)
Sex			0.329	1.482 (0.672–3.264)
male	184 (59.5)	21		
female	125 (40.5)	10		
Age			0.005 **	2.954 (1.377–6.340)
≥55	116 (37.5)	19		
<55	193 (62.5)	12		
BMI			0.987	0.994 (0.468–2.108)
≥27	130 (42.1)	13		
<27	179 (57.9)	18		
Hypertension			0.008 **	2.818 (1.314–6.045)
Yes	119 (38.5)	19		
No	190 (61.5)	12		
Hyperlipidemia			0.007 **	2.804 (1.322–5.950)
Yes	101 (32.7)	17		
No	208 (67.3)	14		
Diabetes			0.428	1.471 (0.567–3.815)
Yes	45 (14.6)	6		
No	264 (85.4)	25		
Smoke			0.508 0.943	1.543 (0.427–5.579) 0.926 (0.114–7.513)
Current smoker	21 (6.8)	3		
Ex-smoker	11 (3.6)	1		
Never smoker	277 (89.6)	27		
Alcohol			0.408	0.656 (0.242–1.779)
Yes	68 (22.0)	5		
No	241 (78.0)	26		
Framingham Risk Score			0.041 *	2.340 (1.035–5.291)
Intermediate-risk level (10–20%)	252 (81.6)	21		
High-risk level (>20%)	57 (18.4)	10		

* *p* < 0.05, ** *p* < 0.01.

**Table 3 ijerph-18-05462-t003:** Multivariate analysis of risk factors for significant coronary artery stenosis.

Risk Factor	Coefficient	SE	OR (95% CI)	*p*
Age ≥ 55 (y/o)	0.999	0.400	2.716 (1.239–5.954)	0.013 *
Hypertension	0.827	0.401	2.287 (1.042–5.019)	0.039 *
Hyperlipidemia	0.969	0.395	2.635 (1.215–5.713)	0.014 *

* *p* < 0.05.

## Data Availability

The data and materials used in this study are available from the corresponding author on reasonable request.

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
