# Peer review of "Use of Coronary Computed Tomography Angiography to Screen Hospital Employees with Cardiovascular Risk Factors"

_ijerph, 2021, doi:10.3390/ijerph18105462_

Round 1

Reviewer 1 Report

The resubmitted manuscript was improved compared to the previous version.

Author Response

We thank the reviewers’ valuable and constructive comments!

Reviewer 2 Report

The Authors have responded to all the observation made. My only further suggestion is to add the use of Kolmogorov-Smirnov test to the description of the statistical analysis. 

Author Response

Comments: The Authors have responded to all the observation made. My only further suggestion is to add the use of Kolmogorov-Smirnov test to the description of the statistical analysis.

Response: According to the reviewer’s comment, we have added the following subsection to the methods. (line 112-114; method part, paragraph 7, row 4-6)

“The data were verified by the Kolmogorov-Smirnov test, which is a normal distribution or satisfies the sample size guidelines for non-normal data prior to parametric statistical analysis.”

Reviewer 3 Report

Applying a method using ionizing radiation for screening purposes can be justified only by a sufficiently high risk of future negative health effects in  patients, in this case myocardial infarction. In the material studied, the vast majority of patients (over 80%) had this risk below 20%. As the authors rightly point out in response to my review, the use of CCTA in asymptomatic patients is controversial. If so, one may even wonder if the authors had the right to perform screening in this form. The fact that they had consents of the bioethics committee and patients does not change it. For this reason, this issue should not be left without comment in this paper. I expect the authors include a paragraph on this issue in the discussion and explain, why they had decided to apply a method using ionizaing radiation in patients with a very low risk of major adverse cardiac events.

Author Response

Comments: Applying a method using ionizing radiation for screening purposes can be justified only by a sufficiently high risk of future negative health effects in patients, in this case myocardial infarction. In the material studied, the vast majority of patients (over 80%) had this risk below 20%. As the authors rightly point out in response to my review, the use of CCTA in asymptomatic patients is controversial. If so, one may even wonder if the authors had the right to perform screening in this form. The fact that they had consents of the bioethics committee and patients does not change it. For this reason, this issue should not be left without comment in this paper. I expect the authors include a paragraph on this issue in the discussion and explain, why they had decided to apply a method using ionizaing radiation in patients with a very low risk of major adverse cardiac events.

Response: According to the reviewer’s comment, we have added the following subsection to the discussion. (line 206-240; discussion part, paragraph 6-7)

“Regarding the use of imaging screen for asymptomatic coronary artery disease remains controversial in both American College of Cardiology/ American Heart Association (ACC/AHA) and European Society of Cardiology (ESC) guidelines. According to ACC/AHA guideline, the coronary artery calcium score is considered an appropriate option for asymptomatic patients as an initial evaluation with an estimated >5% 10-year risk, diabetes mellitus, familial hyperlipidemia, or family history of early coronary artery disease; the CCTA is suitable for symptomatic patients as an initial evaluation without any history of coronary artery disease. According to ESC guideline, the calcium score may be considered as a risk modifier in the cardiovascular disease risk assessment of asymptomatic patients at moderate risk (IIb); screening of coronary artery disease with CCTA may be considered (IIb) [1]. This is a concern with radiation exposure that is higher with CCTA than the coronary artery calcium score. However, with recently introduced new CT-technologies, the radiation dose for CCTA has been significantly reduced. We adapted the newer CT scanner and 40% of adaptive statistical iterative reconstruction, following the rule of as low as reasonably achievable, that the radiation dose was reduced by approximately 30% compared to conventional CT. This makes the CCTA more advantageous, by significantly reducing the difference of radiation dose between CCTA and the coronary artery calcium score, which can offer detailed imaging to provide better analysis and prognostic information. Some studies have demonstrated the benefits of CCTA in asymptomatic patients, including diabetes mellitus, non-calcified plaque, or even young adults [2-4], while the coronary artery calcium score is considered an appropriate option by the guidelines.

This study was designed because two asymptomatic senior supervisors in our hospital, who were classified to intermediate risk level by Framingham Risk Score, both experienced a new cardiovascular event during the same year; and because hospital employees are at greater risk of long working hours, job strain, and shift work, which increases the risk of coronary heart disease. The coronary artery calcium score is a measurement of the amount of calcium in the walls of the arteries that supple the heart muscle, which shows a significant association with the medium- or long-term occurrence of major cardiovascular events. However, the higher the coronary artery calcium score it is, may not represent the real plaque and stenosis of the coronary artery. Regarding coronary artery stenosis, which was evaluated by CCTA, does not mean that the patient will have a heart attack, but this report still represents the actual stenosis of the coronary artery. We therefore designed this study that CCTA was offered for those whose Framingham Risk Score indicates an intermediate or high risk level.”

Reference:

[1] 2019 ESC Guidelines on Diabetes, Pre-diabetes and Cardiovascular Diseases Developed in Collaboration with the EASD - American College of Cardiology Available online: https://www.acc.org/latest-in-cardiology/articles/2020/03/09/13/11/2019-esc-guidelines-on-diabetes-pre-diabetes-and-cvd (accessed on May 10, 2021)

[2] Plank, F.; Friedrich, G.; Dichtl, W.; Klauser, A.; Jaschke, W.; Franz, W.-M.; Feuchtner, G.The diagnostic

and prognostic value of coronary CT angiography in asymptomatic high-risk patients: a cohort study.,

doi:10.1136/openhrt-2014-000096.

[3] Guaricci, A.I.; DeSantis, D.; Carbone, M.; Muscogiuri, G.; Guglielmo, M.; Baggiano, A.; Serviddio, G.;

Pontone, G.Coronary atherosclerosis assessment by coronary CT angiography in asymptomatic diabetic

population: A critical systematic review of the literature and future perspectives. Biomed Res. Int. 2018, 2018.

[4] Jin, K.N.; Chun, E.J.; Lee, C.H.; Kim, J.A.; Lee, M.S.; Choi, S.IlSubclinical coronary atherosclerosis in

young adults: Prevalence, characteristics, predictors with coronary computed tomography angiography. Int. J.

Cardiovasc. Imaging 2012, 28, 93–100, doi:10.1007/s10554-012-0143-0.

This manuscript is a resubmission of an earlier submission. The following is a list of the peer review reports and author responses from that submission.

Round 1

Reviewer 1 Report

The authors in this manuscript aimed to investigate the risk factors for a CAD-RADS score in a large study population composed of hospital employees.
Despite the large study population, several points should be addressed:

1)The authors are invited to define clearly the inclusion criteria by giving a clear definition for dyslipidemia and hypertension. I expect that the recent definitions for such risk factors were done according to the recent updated ESC recommendations.

2) What is meant in Table 1 by `heart diseases`.  When talking about relationship between heart- CT and cardiovascular risk factors then the reader expect a clear definition for the affirmation (heart diseases).

3) Classification for diabetes mellitus (Typ 1 or 2 ) can be such study included

4) Why didn’t the authors  show some important values as HbAc %, proBnp , GFR . 

Author Response

(The authors gave the same response as above.)

Reviewer 2 Report

The paper by Li et al. describes the application of coronary computed tomography angiography (CCTA) in a group of 309 hospital workers with an intermediate or high cardiovascular risk, assessed by the Framingham Risk Score. The results showed that only age ≥ 55 years, hypertension and hyperlipidemia were significant risk factors for the outcome of coronary stenosis, assessed by CAD-RADS.

The design of the study and the methods are clearly described and the results are adequately discussed. However, the limitations of the study, correctly reported by the Authors, make the results not very innovative and limited for the conclusions. As note, the lack of an analysis including the occupational variables is an essential limit, considering that the study is targeted on a specific working population. Moreover, to consider smoking habits only as yes or no could also heavily influence the results observed. Finally, some changes have been suggested to improve some parts of the manuscript.

Suggestions to the Author:

  • Introduction: Previous experience of screening by CCTA in specific groups of workers should be reported to have a clear information on the rational of the study.
  • Methods: The risk condition for Framingham study should be reported also in this section.
  • Methods: Parametric statistical analysis were used, but the methods section does not report if the normality of the data distribution has been previously checked.
  • Results: In the statistical analysis smoking habits should be performed considering separately ex-smokers from no-smokers.
  • Table 2: The Authors should explain the basis for the cutoff used for age (55 years) and BMI (27 years).

Author Response

(The authors gave the same response as above.)

Reviewer 3 Report

The authors screened hospital staff using coronary computer tomography angiography (CCTA)  to identify those who might have serious cardiac problems due to coronary artery disease in the near future. To this end, they used the Framingham scale that allowed  to calculate the risk of serious consequences of coronary artery disease in the next 10 years. In patients with a risk higher than 10%, CCTA was applied, and then, using the standardized method of Coronary Artery Disease Reporting and Data System, the results of the study were divided into categories and further treatment resulted directly from this division
My doubts are raised by the use of the radiological method without paying attention to the exposure of patients to ionizing radiation.According to guidelines from the European Society of Cardiology and the American College of Cardiology/American Heart Association (ACC/AHA), coronary artery calcium score  is considered an acceptable option  for primary screening of asymptomatic patients with Framingham intermediate risk (10–20%). So my question is why authors did not use this imaging modality for sannning patients with this range of risk? It should be stressed that CACS is a source of much lower effective dose (mean value 0.5 mSv) than CCTA (mean values from 2.7 to 4.9 mSv, depending on patient BMI). Dose reduction to patients undergoing studies using ionizing radiation is nowadays an important consideration and should be applied whenever it is possibile. Authors are asked to comment on this.

Author Response

(The authors gave the same response as above.)
